# Progress of Studies on Plant-Derived Polysaccharides Affecting Intestinal Barrier Function in Poultry

**DOI:** 10.3390/ani12223205

**Published:** 2022-11-18

**Authors:** Shiwei Guo, Yuanyuan Xing, Yuanqing Xu, Xiao Jin, Sumei Yan, Binlin Shi

**Affiliations:** College of Animal Science, Inner Mongolia Agricultural University, Hohhot 010018, China

**Keywords:** chemical barrier, immune barrier, microbial barrier, physical barrier, plant-derived polysaccharides, poultry

## Abstract

**Simple Summary:**

Recently, plant-derived polysaccharides have been in general use in animal production, due to their strong hypolipidemic, hypoglycemic, antioxidant, antitumor, anticoagulant, anti-inflammatory, and immunomodulatory activities. The available literature suggests that plant-derived polysaccharides improved performance and health in poultry. Therefore, this review aims to provide a reference for the scientific application of plant-derived polysaccharides in poultry production, while also exploring their effects on the gut microbiome, intestinal permeability, intestinal morphology and immune function, etc.; the potential action mechanisms of plant-derived polysaccharides are briefly summarized.

**Abstract:**

As natural bioactive components, plant-derived polysaccharides have many biological functions, such as anti-inflammatory, antioxidant, anticoccidial, and immunity regulation, and have been widely used in poultry production. In this review paper, firstly, the sources and structures of plant-derived polysaccharides are reviewed; secondly, the effects of plant-derived polysaccharides on the intestinal microbiome, permeability, morphology and immune function of poultry are summarized; thirdly, the potential molecular regulation mechanism of plant-derived polysaccharides on the intestinal barrier function of poultry was preliminarily analyzed. The review paper will bring a basis for the scientific utilization of plant-derived polysaccharides in the poultry industry.

## 1. Introduction

Recently, studies have found that polysaccharides extracted from different types of plants are widely important and have a variety of biological functions, such as hypolipidemic, hypoglycemic, antioxidant, anti-tumor, anticoagulant, anti-inflammatory and immunomodulatory activity [1,2,3]. Generally, the polysaccharide is a common and important biological macromolecule in plants, being safe and having low toxicity [2]. There are many extraction methods for plant-derived polysaccharides, including hot water extraction, enzymatic extraction, ethanol precipitation, microwave-assisted extraction, supercritical fluid extraction, et cetera [4,5]. Different extraction processes have a great influence on the content of polysaccharides, the ratio of monosaccharides and their effects. In the field of polysaccharide research, herbal polysaccharides have been widely studied all over the world, mainly in the field of medicine [6]. However, for the application study in poultry production, we also found that plant-derived polysaccharides improved growth performance [7], promoted nutrient digestion and absorption [8], improved meat quality [9] and, in addition, had antiviral [10,11], anti-coccidiosis [12] and anti-tumor activity [13], enhanced immunity given by vaccines [7], regulated immune and antioxidant functions [14], and improved intestinal barrier function [15].

Studies have found that plant-derived polysaccharides could maintain the integrity of the intestinal barrier, reduce the level of intestinal inflammation, repair damaged intestinal tissue, restore normal intestinal structure, and assist the digestion and absorption of nutrients in poultry [8,16,17]. The small intestine is considered an important digestive and absorption organ in poultry, and is the largest immune organ. The intestinal barrier mainly includes physical, chemical, immune and microbial barriers, and plays an important role in defense against pathogens, antigens, toxins and other harmful substances. The physical barrier consists of the mucosal layer, intestinal epithelial cells and tight junctions, which play an important role in regulating intestinal permeability. The chemical barrier consists of digestive enzymes, glycoprotein mucin and bacteriostatic substances, which can prevent intestinal epithelial cell damage and bacterial translocation. The immune barrier refers to the lymphatic tissue associated with the intestinal tract and the humoral immune factors secreted, such as immunoglobulins, complements and cytokines, which play an important role in preventing pathogen invasion. The symbiotic microorganisms in the gut constitute an intestinal microbial barrier, which not only inhibits the proliferation of pathogens by competing for limited nutrients and releasing antimicrobial substances, but also provides the host with a variety of physiological processes [18]. At present, most of the published reviews are about the protective effects of plant-derived polysaccharides on intestinal microecology, intestinal microbiota and intestinal health of animals [19,20,21,22], however, this is not specific to poultry, so this review mainly summarizes the role of phytogenic polysaccharides on the gut barrier in poultry; in addition, the effects of plant-derived polysaccharides on intestinal physical, chemical, immune and microbial barriers of poultry are systematically summarized.

## 2. Source and Structure of Plant-Derived Polysaccharides

The main sources of plant-derived polysaccharides applied in poultry production are shown in Table 1. From Table 1, we can intuitively see the plants as the source of plant-derived polysaccharides, application forms, experiment objects, and their main functions in poultry production. In terms of polysaccharide structure, different types of monosaccharides from plant-derived polysaccharides have different linking sites between different monosaccharides—glycosidic bonds [23]. The composition of plant-derived polysaccharides mainly includes glucan, mannan, pectin polysaccharide, arabinogalactan and galactan [2,24,25].

## 3. Effects of Plant-Derived Polysaccharides on Intestinal Barrier Function in Poultry

### 3.1. Plant-Derived Polysaccharides Improve Intestinal Microbial Barrier in Poultry

The gut microbiome barrier is composed of intestinal microbiota that adhere to the intestinal mucosa layer. Intestinal microbiota and tact form a dynamic and stable micro-ecosystem [53]. At present, the studies on the effects of plant-derived polysaccharides on intestinal microbe of poultry mainly focus on the jejunum, ileum and cecum. It has been reported that 100% microbially fermented feed and ginseng polysaccharide (FP) and 100% complete feed and ginseng polysaccharide (Po) significantly increased the abundance of *Sutterella* and decreased the abundance of *Asteroleplasma* in jejunum content of Xuefeng black-bone chickens at the genus level [41]; moreover, at the species level, the FP group significantly increased the abundance of *Bacteroides_vulgatus* and *Eubacterium_tortuosum* and decreased that of *Mycoplasma_gallinarum* and *Asteroleplasma_anaerobium*, while the Po group significantly increased *Mycoplasma_gallinarum* and *Asteroleplasma_anaerobium* abundance [41]. Another study found that dietary supplementation consisting of 1 g/kg *Acanthopanax senticosus* polysaccharides increased *Lactobacillus* counts and decreased *Escherichia coli* and *Salmonella* counts in ileal contents of broilers compared with the control diet [3], and Ao et al. [9] also found that dietary supplementation of *Achyranthes bidentata* polysaccharides (0.02–0.04%) has similar effects on Pekin ducks. It is worth noting that most researchers focus on the effects of plant-derived polysaccharides on the cecal microorganisms of poultry. Wang et al. [42] reported that adding more than 200 mg/kg *Camellia oleifera* cake polysaccharides to the diet of Lingnan yellow broilers increased the number of *Lactobacillus* and *Enterococcus faecalis* in the cecum compared to the control, and had the potential to inhibit the growth of *Escherichia coli*. Furthermore, Xiang et al. [43] found that dietary Yingshan Yunwu tea polysaccharides (200, 400, 800 mg/kg) enhanced *Bacteroidetes* and *Lactobacillus* abundance, and reduced that of *Proteobacteria* in the cecum of broilers compared to the normal control group.

Except for the above normal feeding conditions, plant-derived polysaccharides could also play a role in poultry under challenging conditions such as pathogen growth. When broilers are infected with *Salmonella serotype* resulting in enteritis, a dietary addition of 500 mg/kg alfalfa polysaccharides enhanced the abundance of beneficial microorganisms in the intestinal tract, such as *Bacteroides*, *Barnesiella*, *Parabacteroides*, *Butyricimonas*, *Prevotellaceae* while decreasing facultative anaerobic bacteria, including *Proteobacteria*, *Actinomycetes*, *Ruminococcaceae*, *Lachnospiraceae* and *Burkholdeaceae* [34]. Moreover, Cui et al. [35] found that dietary Caulis Spatholobi polysaccharide (0.2, 0.4, and 0.6%) increased cecum species richness, restored cyclophosphamide-induced intestinal microbiota imbalance, and improved *Lactobacillus* abundance. Liu et al. [54] also reported that dietary supplementation of γ-irradiation *Astragalus* polysaccharides increased the number of operational taxonomic units (OTUs) in cecal contents induced by cyclophosphamide, suggesting that polysaccharides could enrich and diversify the bacterial community; this is mainly reflected in that polysaccharides reduced the abundance of *Faecalibacterium*, *Bacteroides* and *Butyricicoccus* and improved the proportion of *Ruminococcaceae* UCG-014, *Negativibacillus*, *Shuttleworthia*, *Sellimonas* and *Mollicutes* RF39_norank. The appropriate supplemental level of polysaccharide was 900 mg/kg. Therefore, plant-derived polysaccharides can improve the diversity of microflora, regulate the structure and composition of the microflora, and restore the gut microbial balance.

One of the reasons why plant-derived polysaccharides regulate intestinal microbiota is closely related to short-chain fatty acids (SCFA). Short-chain fatty acids (SCFAs) can stimulate the proliferation and differentiation of intestinal epithelial cells and increase villus height, thus increasing the absorption surface area. SCFAs are metabolites produced by bacterial fermentation of feed fiber in the hindgut and are mainly composed of acetic acid, propionic acid and butyric acid. SCFAs (mainly butyric acid) consume oxygen in the gut, creating an anaerobic environment that reduces the expansion of aerobic pathogens such as *Salmonella* in the gut; butyric acid can be synthesized from proteins via the lysine pathway, suggesting that gut microbiota can adapt to the change in fermentation substrates and maintain metabolite synthesis [55]. The specific metabolic pathways of SCFAs are shown in Figure 1. Song et al. [32] found that dietary 200 ppm *Astragalus* polysaccharides reduced *Clostridium perfringens* counts in cecum, increased *Romboutsia*, *Staphylococcus* and *Halomonas* abundance in the ileum, decreased *Lactobacillus* abundance, improved the ileum microflora, increased the abundance of the SCFA-producing bacterium *Romboutsia*, significantly increased the levels of propionic acid and butyric acid in the ileum, and tended to increase the levels of isobutyric acid and hexanoic acid, which reduced the inflammatory damage caused by necrotic enteritis. In addition, Lv et al. [29] reported that dietary supplementation consisting of 200 mg/kg *Astragalus* polysaccharide increased the relative abundance of *Firmicutes* and *Lactobacilli* in the cecum; in this study, *Firmicutes* and *Bacteroides* were the most abundant bacteria in the cecum, accounting for more than 60% of the total bacteria in Chongren hens, and facilitated energy absorption and conversion of polysaccharides into absorbable monosaccharides and SCFAs. It had been found that dietary supplementation consisting of 900 mg/kg γ-irradiation *Astragalus* polysaccharides increased the concentrations of acetic acid and butyric acid in cecum contents [54]. Similarly, Nguyen et al. [56] also reported higher concentrations of SCFA (propionic acid, isobutyric acid, valeric acid) in the cecum of broilers fed diets supplemented with non-starch polysaccharides (high-dose group).

In general, plant-derived polysaccharides could selectively promote the proliferation of beneficial intestinal microbiota, inhibit the implantation of harmful microbiota, and regulate the structure and metabolic function of intestinal microbiota. However, different sources of polysaccharides have different probiotic effects on intestinal microbiota [57]. Besides, since most plant-derived polysaccharides cannot be digested and degraded in the stomach and small intestine, they are degraded by bacteria in the large intestine and then biotransformed to exert their biological functions [58]. Furthermore, metabolomic analysis of intestinal contents revealed that *Astragalus* polysaccharides might influence butyrate metabolism, alanine, aspartate and glutamate metabolism, the conversion of pentose and glucuronide, and D-glutamine and D-glutamate metabolism [59]. Among other types of polysaccharides, they can serve as their main energy source and can modulate the related metabolic pathways, fermentation metabolites and SCFA production, which in turn act on the gut microbiota; for example, an increase in butyric acid decreases the pH, which in turn inhibits the proliferation of pathogenic bacteria [60]. Potentially, plant-derived polysaccharides can be ingested by the body as a carbon source of intestinal microbiota, providing conditions for the proliferation of beneficial bacteria and the growth and development of intestinal immune organs [22]. Moreover, the abundance of intestinal microbiota is correlated with villus morphology [41]. Plant-derived polysaccharides could also promote the morphological development of intestinal villi (See Section 3.4).

### 3.2. Plant-Derived Polysaccharides Improve Intestinal Chemical Barrier in Poultry

The intestinal chemical barrier is mainly composed of a mucus layer that prevents direct contact with host intestinal tissue cells by altering the location of gut microbes. Some substances produced in the intestine act as chemical barriers, such as gastric acid, bile salts, mucopolysaccharides, digestive enzymes, lysozyme and glycoproteins [61]. It was found that dietary non-starch polysaccharides could reduce cecal pH in broilers [56], and have a certain inhibitory effect on pathogenic bacteria in the cecum. Moreover, secretory immunoglobulin A (sIgA) is mainly distributed in the intestinal mucus layer, which is an important indicator to evaluate the intestinal barrier. For this point, studies on *Astragalus* polysaccharides (APS) are relatively abundant. Yang et al. [28] found that injection of APS (1, 2, and 4 mg APS was dissolved into 0.5 mL normal saline) into eggs improved the content of sIgA in jejunum wash of broilers at different numbers of days after hatching compared to the control group. Besides, Shan et al. [26] found that oral administration of 0.5 mL APS, compared to the non-vaccinated negative control group and vaccinated control group, increased the level of sIgA in the jejunal mucosa of chicks immunized against Newcastle Disease. In addition, oral administration of 0.3 mL (purity 87.81%) APS could also increase the content of sIgA in the jejunum of goslings infected with gosling plague [1]. It was found that adding 0.6 g/L APS to drinking water significantly increased the content of sIgA in the ileum of Muscovy duck reovirus-infected young Muscovy ducks [15]. Except for APS, oral administration of 20 and 40 mg/mL *Pinus massoniana* pollen polysaccharides could also increase the secretion of intestinal sIgA content in broilers [50]. In addition, Wang et al. [48] reported that injection of 10, 20 and 40 mg/mL *Paulownia fortunei* flower polysaccharides into the neck of broilers promoted sIgA secretion in the duodenum caused by cyclophosphamide.

Goblet cells are specialized intestinal epithelial cells that secrete a variety of proteins (Mucin-2, trefoil factor 2) that contribute to maintaining the integrity of intestinal mucosa. It was found that the supplementation of 900 mg/kg γ-irradiated APS significantly increased the number of goblet cells in the three small intestinal segments of broilers induced by cyclophosphamide [30]. Mucin 2 (Muc-2) is an important component of the intestinal mucous layer, secreted by goblet cells, and plays an important role in avoiding intestinal mucosa damage caused by adverse factors, lubricating the intestine to maintain normal barrier function, and participating in nutrient absorption and resistance to the invasion of bacteria, viruses, parasites and enteritis. Muc-2 deficiency leads to diseases directly in contact with the intestinal mucous layer, thereby resulting in spontaneous colitis. It has been found that dietary supplementation with 500 mg/kg alfalfa polysaccharides could increase the gene expression level of Muc-2 in jejunum, and thus improve the intestinal function of *Salmonella* serotype (ser.) Enteritidis-challenged broilers [34]. Similarly, it had been reported that dietary supplementation with 4 mg/kg Mulberry leaf polysaccharide enhanced the expression of Muc-5 in the respiratory tract of chicks immunized with the Newcastle Disease vaccine, and the mechanism is related to the up-regulation of TLR7 levels [46]. Therefore, it is speculated that mucin expression in the intestinal tract is closely related to TLR level.

Previous studies showed that dietary supplementation of plant-derived polysaccharides increased the activity of lipase and amylase in the intestine of broilers, thus enhancing the absorption capacity of nutrients in the small intestine and strengthening the intestinal chemical barrier function. Long et al. [45] found that with the increase in the supplemental level of dietary *Lycium barbarum* polysaccharides, the activity of protease, amylase and lipase increased in the small intestine of broilers, indicating that *Lycium barbarum* polysaccharide supplementation induces the expression of digestive enzymes. Wu [62] found that dietary *Astragalus membranaceus* polysaccharide administration increased the activities of digestive enzymes including amylase, lipase, and protease in juvenile broilers. However, the specific mechanism of the effects of plant-derived polysaccharides on the digestive enzyme activity of poultry is hitherto unclear, and might be related to glucose digestion and absorption in the small intestine. There are three underlying reasons: (1) dietary supplementation polysaccharides could increase the expression of glucose transporter (SGLT1), GLUT2 and GLUT5 genes in the duodenum, jejunum and ileum of broilers, and the expression of these genes might contribute to digestion and absorption in the small intestine [52]. Similar studies found that broilers fed non-starch polysaccharide diets supplemented with wheat and barley significantly increased intestinal α-amylase and lipase activity, and significantly increased gene expression levels of SGLT1 and Muc-2 in jejunum [63]. (2) Polysaccharides can retain water, promote peristalsis, encourage the feeling of satiety, and retard the rate of emptying [20], thus promoting the digestion and absorption of nutrients. (3) The reason why dietary plant-derived polysaccharides positively influence the activity of digestive enzymes may be attributed to their beneficial effects on beneficial bacteria in the intestinal tract and the intestinal morphology of poultry [64].

### 3.3. Plant-Derived Polysaccharides Improve Intestinal Physical Barrier in Poultry

The intestinal physical barrier occupies a central position in the intestinal barrier structure, which is composed of intestinal epithelial cells and intercellular junctions. The junctions between cells include tight junctions, adhesion junctions, desmosomes, etc. Tight junctions are particularly important; they are located at the top of the outer membrane of intestinal epithelial cells and are long and narrow bandlike structures that block the gap between cells, thereby preventing macromolecular substances in the intestinal lumen, such as bacteria and toxins, from entering the blood. In addition, the permeability of tight junctions between cells determines the barrier function of the whole intestinal epithelial cells, which are composed of a variety of proteins with specific functions, including Claudins, Occludin, junction adhesion molecules and intracellular proteins such as zonula occludens (ZO). Claudins and Occludin are the skeletons of the tight junction structure, and ZO is the basis of the tight junction supporting structure to bind transmembrane proteins [65] (Figure 2). The tight junction structure has high dynamic stability, its permeability is regulated by intracellular and extracellular signals, and it is susceptible to diet, disease, stress, et cetera. Li et al. [34] reported that dietary supplementation with 500 mg/kg alfalfa polysaccharides could improve the gene expression of tight junction proteins Claudin-1 and Occludin in the jejunum of broilers, thus alleviating the intestinal challenge response caused by *Salmonella* serotype (ser.) Enteritidis. Furthermore, Wang et al. [30] also found that dietary supplementation with 900 mg/kg γ-irradiation *Astragalus* polysaccharides increased the gene expression of Claudin-1, ZO-1 and Occludin in the jejunum of broilers induced by cyclophosphamide; the effect is better when animals are fed with xylo-oligosaccharides. In addition, Cui et al. [35] also reported that dietary supplementation of 0.2, 0.4, and 0.6% Caulis spatholobi polysaccharides could increase the gene expression of ZO-1 and Claudin-1 in jejunum mucosa, thereby improving intestinal barrier function in cyclophosphamide-induced immunosuppressive chickens. Tight junction proteins are regulated by a variety of intracellular pathways, including myosin light-chain kinase (MLCK), mitogen-activated protein kinase (MAPK), protein kinase C (PKC) and the Rho family of small GTPases. The MLCK pathway is one of the most abundant pathways in the small intestine and is a key step in the regulation of tight junction permeability by multiple external stimuli such as cytokines and pathogens; inhibition of MLCK could prevent the deterioration of barrier function [66]. Currently, there have been reports on the molecular mechanism of plant active ingredients regulating the tight-junction protein in poultry [67]. However, in terms of plant-derived polysaccharides, the specific regulatory mechanism of the intestinal tight-junction protein is rarely reported in poultry.

Intestinal mucosal morphology is an important component of the physical barrier, and is related to the absorption capacity of nutrients. When villus height increases, it means that the number of mature villus cells increases, the area of the small intestine to absorb nutrients increases, and the ability to absorb nutrients is high. The main function of the crypt is to generate cells and continuously replenish villous epithelial cells. The depth of the crypt is closely related to the maturation rate of epithelial cells. The deepening of the crypt indicates that the maturation rate of villous epithelial cells decreases and that their secretion ability is weakened. Villus height, crypt depth and the ratio between them are important indicators to evaluate the morphological and structural integrity and functional status of the intestinal mucosa. A large villus height/crypt depth ratio indicates strong intestinal digestion and absorption capacity, while a small villus height/crypt depth ratio indicates weak intestinal digestion and mucosal damage, affecting digestion and absorption of nutrients.

First, Yang et al. [28] found that by injecting *Astragalus* polysaccharides (1, 2, and 4 mg APS was dissolved into 0.5 mL normal saline) in ovo, compared to the control group, villus height (VH) was increased, crypt depth (CD) was decreased, and villus height/crypt depth (V/C) was increased in the small intestine of broilers at different ages. Liao et al. [15] also found that adding 0.6 g/L *Astragalus* polysaccharides to drinking water could effectively restore VH and improve V/C ratio in the small intestine, thus alleviating small intestine damage in Muscovy ducks infected with Muscovy reovirus. Moreover, Sha et al. [50] reported that oral administration of *Pinus massoniana* pollen polysaccharides at 10–40 mg/mL could effectively promote the healthy development of intestinal villi in broilers, but there was no significant difference between the 20 mg/mL and 40 mg/mL dose groups, suggesting that the promoting effect of *Pinus massoniana* pollen polysaccharides was not always presented in a dose-dependent manner. Except for injection and drinking water, most researchers used the method of mixing plant-derived polysaccharides into poultry basal diets. Lv et al. [29] reported that dietary supplementation with 100–400 mg/kg *Astragalus* polysaccharides increased VH in the duodenum, jejunum and ileum, and CD and V/C in the jejunum of Chongren laying hens compared to the control. Other studies have found that dietary xylo-oligosaccharides and γ-irradiated *Astragalus* polysaccharides had an interactive effect on improving ileal VH and V/C, and the V/C values of the duodenum and jejunum in the xylo-oligosaccharides and γ-irradiated *Astragalus* polysaccharides group are higher than that in the γ-irradiated *Astragalus* polysaccharides group [30]. In addition, Xie et al. [41] found that dietary supplementation with 200 g/t ginseng polysaccharides significantly increased jejunum VH and V/C ratio and decreased CD in Xuefeng black-bone chickens compared to the 100% microbially fermented feed group. Furthermore, plant-derived polysaccharides could effectively alleviate the intestinal mucosal morphological injury caused by external factors in poultry. It has been found that dietary 500 mg/kg alfalfa polysaccharides significantly increased the VH and V/C ratios of duodenum and jejunum of broilers and reduced CD, thus alleviating intestinal injury caused by *Salmonella* serotype (ser.) Enteritidis [34]. Cui et al. [35] also reported that dietary 0.2, 0.4, and 0.6% Caulis spatholobi polysaccharides increased intestinal V/C, and reduced intestinal injury induced by cyclophosphamide in broilers. Furthermore, dietary supplementation with 200 ppm *Astragalus* polysaccharides could reduce the intestinal lesion score and pathological score, increase jejunum V/C ratio and decrease CD, and it is speculated that *Astragalus* polysaccharides could reduce necrotic enteritis-induced intestinal injury and prevent the invasion of pathogenic microorganisms in the blood, thus alleviating systemic immune response [32].

Intestinal permeability is one of the indicators that indirectly reflect the damage to the intestinal mucosal physical barrier. Diamine oxidase (DAO) is a key intracellular enzyme in the villus cells of the upper mucosa and is an important indicator for monitoring intestinal barrier function. When the intestinal mucosa is damaged, the intracellular release of DAO increases, which enters the blood to increase DAO in plasma, and the DAO activity of intestinal mucosa increases. Moreover, serum endotoxin (ET) and D-lactic acid (D-LA) reflect intestinal permeability. When intestinal permeability increases, the content of ET and D-LA in serum will significantly increase, and when the intestinal barrier is damaged or weakened, intestinal permeability increases. The study found that dietary supplementation with γ-irradiated *Astragalus* polysaccharides decreased plasma D-LA concentration in broilers [30]. Moreover, dietary supplementation with 300 mg/kg *Astragalus* polysaccharides or 150 mg/kg *Glycyrrhiza* polysaccharides reduced serum DAO activity [17]. However, the dietary inclusion of 500 mg/kg alfalfa polysaccharides increased DAO content in the duodenum and jejunum of broilers [34]. Moreover, by combining the pharmacokinetic analysis and cell monolayer transport mechanisms of lentinan, it may be speculated that polysaccharides can be absorbed and circulated in the blood through macropinocytosis and clathrin-mediated endocytosis [68], and then regulate the concentration of permeability indicators in the blood. To sum up, it can be preliminarily speculated that plant-derived polysaccharides may maintain intestinal barrier integrity and ensure normal barrier function by improving intestinal morphology, upregulating tight-junction proteins of the small intestine, and decreasing intestinal permeability [17,19,35].

### 3.4. Plant-Derived Polysaccharides Improve the Intestinal Immune Barrier in Poultry

The gut generally acts as an organ for digestion and absorption, but is also an important immune organ, containing approximately 70% of the body’s total immune cells [69]. In the intestinal immune barrier, intestine-associated lymphoid tissue plays an important role, among which lymphocytes and macrophages play a crucial role in the process of resistance to pathogen invasion. It has been reported that oral administration of 10, 20, and 40 mg/mL *Pinus massoniana* pollen polysaccharides to broilers, compared with the PBS group, promoted the transformation rate of CD4+ and CD8+ T lymphocytes and the gene expression of intestinal mucosal cytokines IL-2, IL-4 and IFN-γ [50]. Liao et al. [15] found that the supplementation with 0.6 g/L *Astragalus* polysaccharides in drinking water effectively inhibited the reduction of intraepithelial lymphocytes and goblet cells caused by reovirus infection in Muscovy ducks, increased the secretion of IL-15, and brought the levels of TNF-α and IFN-γ to normal levels, thus alleviating the damage of the immune barrier in the small intestinal mucosa of Muscovy ducklings and effectively improving the immune function of the intestinal mucosa.

The intestinal immune defense system is mainly composed of intestine-associated lymphoid tissue existing in the intestinal wall and its secreted IgA, IgM, and IgE. Shan et al. [26] found that oral administration of 0.5 mL (1, 2 and 4 mg/mL) *Astragalus* polysaccharides to broilers promoted the growth of IgA+ cells in jejunum and facilitated the secretion of sIgA, thereby ameliorating the jejunal immune function of chickens vaccinated against Newcastle disease. In addition, the study found that dietary supplementation with 500 mg/kg *Achyranthes bidentata* polysaccharides reduced the levels of S-IgA and TNF-α in jejunum mucosa of broilers. The gene expression of NF-κB in the jejunum of *Achyranthes bidentata* polysaccharides group was lower than that of the basal diet group and *Escherichia coli* K88 challenge group; furthermore, it was found that polysaccharides regulated IL-4 and TNF-α contents and the production of pro-inflammatory mediators through NF-κB signaling pathway, thereby improving the immune function of *Escherichia coli* K88 challenge broilers [40]. Li et al. [31] study found that compared with the cyclophosphamide challenge group, dietary supplementation with *Astragalus* polysaccharides and γ-irradiated *Astragalus* polysaccharides improved the jejunal IL-2, IL-10 and IFN-γ gene expression level, suggesting that polysaccharides could moderately activate the intestinal innate immune response mechanism, improve the immune function mediated by intestinal mucosa cells, and relieve the immunosuppressive response induced by cyclophosphamide. In particular the number of jejunal goblet cells as well as IL-2 and IL-10 gene expression in the 900 mg/kg γ-irradiated *Astragalus* polysaccharides group were higher than those in the 900 mg/kg *Astragalus* polysaccharides group. Moreover, oral administration of *Epimedium* polysaccharides to broilers in vivo could promote the titer of anticoagulant antibodies, enhance the proliferation of peripheral blood lymphocytes, and promote the release of cytokines in the blood (IFN-γ, IL-4 and IL-2) and duodenum (IL-17). Oral administration of *Epimedium* polysaccharides given three times also significantly improved the immune capacity of duodenal mucosal cells [70]. Song et al. [32] reported that dietary 200 ppm *Astragalus* polysaccharides reduced the levels of RORα, IL-17F and IL-17F/IL-10 in the intestinal tract, increased the proliferation activity of ileal T and B cells, increased the IL-4 mRNA level in ileum, decreased the proportion of Th17 and Th17/Treg cells in ileum of broilers, and increased ZO-1, thereby alleviating intestinal inflammation caused by necrotic enteritis.

A study found that injecting a certain amount of *Astragalus* polysaccharides in ovo at the early stage of incubation promoted early intestinal development and mucosal immunity by improving cytokine IL-2, IL-4, IFN-γ mRNA expression and Toll-like receptor (TLR)-4 genes in the ileum of chicks after hatching, providing timely and effective protection for chicks [28]. Luo et al. [1] showed that intramuscular injection of 0.3 mL (purity 87.81%) *Astragalus* polysaccharides reduced the gene expression of IL-1β, IL-6 and TNF-α in jejunum of seven-day-old goslings, inhibited the gene expression of NF-κB, COX-2 and PGE2 in jejunum, and effectively reduced intestinal inflammation and injury in goslings infected with disease; its mechanism might be closely related to the NF-κB pathway. Moreover, Cui et al. [35] found that dietary supplementation with 0.2, 0.4, and 0.6% Caulis spatholobi polysaccharides enhanced TLR4, MyD88, NF-κB, Claudin1 and ZO-1 mRNA expression by activating the TLR/MyD88/NF-κB pathway, thereby alleviating cyclophosphamide-induced immunosuppression to protect the intestinal tract of broilers. TLRs are widely distributed in intestinal immune cells and play an important role in the innate and adaptive intestinal immune system. Previous studies have found that the immune regulation action of polysaccharides is related to TLRs, and is closely related to TLR4. TLR4 is one of the earliest innate immune receptors to mediate the activation of the NF-κB and MAPK signaling pathways.

TLR4 receptor proteins bind to corresponding ligands to activate MAPK family proteins (JNK, p38 and extracellular signal-regulated kinase 1/2 (ERK1/2)), which ultimately activate NF-κB downstream signaling pathways to induce and promote the production of inflammatory cytokines such as TNF-α and IL-1β in the intestinal tract [69]. Furthermore, Su et al. [27] showed that oral administration of 15 mg *Astragalus* polysaccharides to broilers significantly inhibited the gene expression of TLR4 and NF-κB in chickens challenged by *Escherichia coli* infection, inhibited intestinal microvascular injury induced by *Escherichia coli* through the regulatory pathway of TLR4-NF-κB, significantly decreased the mRNA levels of inflammatory factors TNF-α, IL-1β and inducible nitric oxide synthase (iNOS), decreased the gene expression levels of vascular adhesion molecule E-selectin, intercellular adhesion molecule-1 (ICAM1) and vascular cell adhesion molecule-1 (VCAM-1), and attenuated the mRNA expression of epithelial growth factor (EGF) and basic fibroblast growth factor (bFGF). It has also been reported that dietary *Astragalus* polysaccharides changed the immune microenvironment of jejunum mucosal epithelial cells and intestinal mucosal immune cells. The adaptability of the jejunum mucosal epithelial cells and intestinal mucosal immune cells appeared after 44 weeks of feeding, and the changes in the immune microenvironment significantly reduced the DNA methylation modification of SOCS1 promoter region in jejunum mucosa of male breeders. This paternal regulation effect affects the histone modification of MyD88 and TRIF in the offspring jejunum mucosa and activates class I interferon and the downstream SOCS1-related endotoxin tolerance signaling pathway. The regulation effect of paternal *Astragalus* polysaccharides could activate the endotoxin-tolerant response state of jejunum mucosa of broilers, while the 10 g/kg *Astragalus* polysaccharide group has a better effect [71]. In addition, the potential mechanisms of PI3K-Akt signaling, Ras signaling, JAK-STAT signaling, NF-κB signaling and estrogen signaling may be involved in regulating the intestinal immune barrier of poultry using plant-derived polysaccharides. Wang et al. [13] found that after feeding broilers with 5 mg Taishan *Pinus massoniana* pollen polysaccharides orally for 7 consecutive days and then injecting Fu-J (SDAU1005) for toxication, polysaccharides were involved in the regulation of the PI3K-Akt signaling pathway, Ras signaling pathway, JAK-STAT signaling pathway, NF-κB signaling pathway, and estrogen signaling pathway. It has been speculated that Taishan *Pinus massoniana* pollen polysaccharides might inhibit Fu-J replication and tumor growth by regulating these signaling pathways. Besides, in vivo experiments also found that injection of *Paulownia fortunei* flowers polysaccharides into the neck enhanced the cellular and humoral immune function of chickens. In addition, monosaccharide analysis showed that *Paulownia fortunei* flowers polysaccharides were mainly composed of galactose, rhamnose, glucose, and arabinose, which might be related to the immunomodulatory properties of plant-derived polysaccharides [48]. Furthermore, numerous studies have shown that the structural characteristics of plant polysaccharides, such as molecular weight, chemical composition, branching structure, and conformation, affect their biological activity [21].

For the molecular regulation mechanism of plant-derived polysaccharides in regulating the intestinal immune barrier of poultry, there are some in vitro experiment reports that may be referenced. Yang et al. [51] reported that polysaccharides from Taishan *Pinus massoniana* pollen had a significant immune stimulation effect on peripheral blood lymphocytes of chickens, mainly manifested by the secretion of IL-2, IL-6 and IFN-γ increasing with the increase in polysaccharide levels, especially 100 μg/mL. In addition, the functional analysis of differentially expressed proteins in GO, KEGG, and COG databases revealed that Taishan *Pinus massoniana* pollen polysaccharides might mediate innate host responses through the TLR and NLR signaling pathways. The intestinal tract has many lymphocytes, and the mechanism by which polysaccharides regulate immune function may be related to T lymphocytes. A previous study found that dietary supplementation with 400 mg (kg body weight) *Atractylodes macrocephala* Koidz polysaccharides alleviated the decrease in the activation level of T lymphocytes challenged by cyclophosphamide through the novel_mir2/CTLA4/CD28/AP-1 signaling pathway [38], and restored the body to normal immune levels. Our previous in vitro study found that 25, 50, 100, and 200 μg/mL *Artemisia argyi* polysaccharides specifically promoted the production of immunoglobulins (IgM and IgG) and cytokines (IL-1β, IL-6 and TNF-α) in the supernatant of peripheral blood cells, and promoted the production of NO in vitro, but not by activating the TLR4/NF-κB signaling pathway [72].

Furthermore, plant-derived polysaccharides may regulate the immune barrier by regulating the intestinal antioxidant status. Yue et al. [49] found that 1, 10 and 100 mg/mL selenizing *Schisandra chinensis* polysaccharides significantly enhanced the activity of antioxidant enzymes SOD, CAT and GSH-Px, and significantly protected liver cells from oxidative damage induced by H_2_O_2_. This mechanism occurred through the regulation of MAPKs and mitochondria-dependent protein expressions of Bcl-2, p-JNK1, p-ERK1/2, p-p38, Bax, caspase 3, and cytochrome C in apoptosis signaling pathways. In addition, 40 mg/L *Astragalus* polysaccharides significantly reduced cadmium-induced ROS generation and expression of LC3-II and Beclin-1 protein in chicken embryo fibroblasts, increased the expression of mTOR and antioxidant levels, and restored viability and morphological damage in chicken embryo fibroblasts [33]. Moreover, 200 μg/mL of *Platycodon grandifloras* polysaccharides also attenuated chromium-induced mitochondrial damage by regulating ROS and MMP, specifically in alleviating chromium-induced PINK1/parkin-mediated mitochondrial autophagy in embryo fibroblasts cells [44]. Overall, the results of these in vitro studies provide a reliable theoretical basis for future research on the molecular mechanism of plant-derived polysaccharides regulating intestinal barrier function in poultry.

## 4. Conclusions and Perspectives

In this review, the sources of plant-derived polysaccharides used in poultry production were described, and the effects of plant-derived polysaccharides on intestinal microbial, chemical, physical, and immune barriers were introduced. In terms of microbial barriers, plant-derived polysaccharides regulated the abundance of beneficial bacteria and harmful bacteria, and could be regulated by the content of short-chain fatty acids. In terms of chemical and physical barriers, plant-derived polysaccharides promoted mucin, digestive enzymes and villus morphology, and decreased intestinal permeability. In terms of the immune barrier, plant-derived polysaccharides regulated immunoglobulins, cytokines, and cellular mediators in the intestinal tract, and regulated them through the corresponding signaling pathways. Whether plant-derived polysaccharides play a role may be related to the species of poultry and whether the body is in a stress state. This review can provide a reference for the study on the effects of plant-derived polysaccharides on the intestinal barrier in poultry. In addition, it can be seen from the above-mentioned reports of scholars that plant-derived polysaccharides are widely used in poultry intestines, and in the future, there will be some breakthroughs as follows: on the one hand, the chemical characteristics of polysaccharides, such as monosaccharide composition, glycosidic bond type, backbone, molecular weight, and surface morphology, will be understood, which will help to explore the mechanism of protecting the intestinal barrier in poultry; on the other hand, RNA sequencing technology can be used to study the underlying mechanisms of plant-derived polysaccharides protecting the intestinal barrier function in poultry. Therefore, plant-derived polysaccharides will be a potential research object with promising prospects.

## Figures and Tables

**Figure 1 animals-12-03205-f001:**
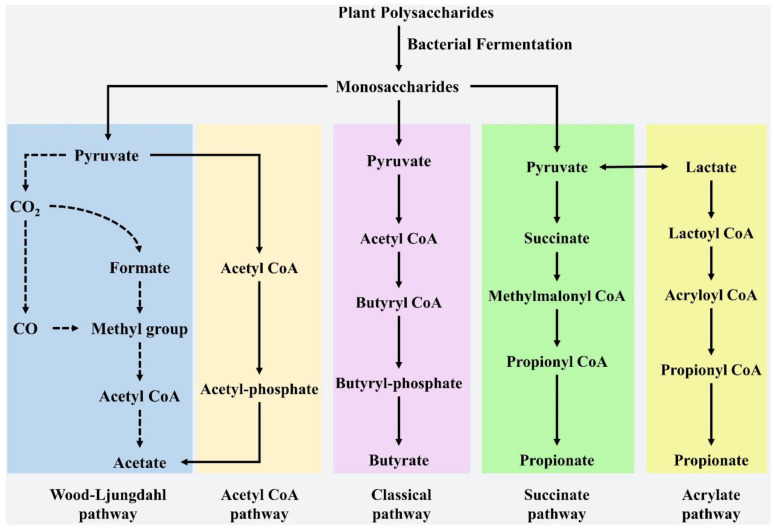
Synthesis pathway of SCFAs [55] Reprinted with permission from Ref. [55], Liu et al. (2021).

**Figure 2 animals-12-03205-f002:**
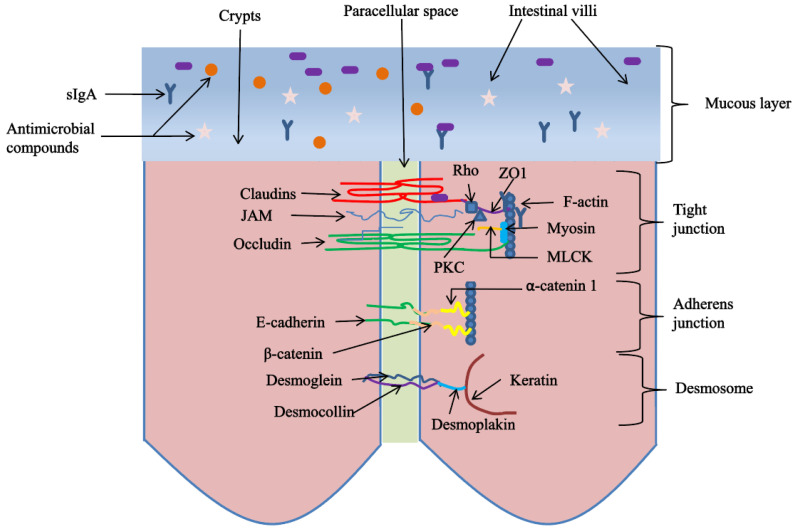
Schematic diagram of intestinal epithelial intercellular junction complex and intestinal mucosal layer [67] Reprinted with permission from Ref. [67], Patra. (2020).

**Table 1 animals-12-03205-t001:** Sources of plant-derived polysaccharides applied in poultry production.

Source	Application Form	Experiment Object	Main Function	Reference
*Astragalus* polysaccharide, 70% content	At dosage of 0.6 g/L in drinking water	Muscovy ducklings	Improved intestinal mucosal immune function and morphology	[15]
*Astragalus* polysaccharides, net content 70%	Orally gavaged daily with 0.5 mL (at doses of 1, 2, 4 mg/mL for four consecutive days)	Hy-Line male chickens	Enhanced the jejunum mucosal immune function	[26]
*Astragalus membranaceus* polysaccharide	Oral 1.5 mL daily (10 mg/mL)	Male Gushi chickens	Alleviated intestinal inflammatory processes and vascular dysfunction	[27]
*Astragalus* polysaccharide	Injected with 0.5 mL of 3 different concentrations solution (1, 2, 4 mg in 0.5 mL physiological saline)	Fertilized eggs	Promoted intestinal development and mucosal immunity	[28]
*Astragalus membranaceus* polysaccharide, purity 87.81%	Intramuscular injected with 0.3 mL	Goslings	Enhanced intestinal antioxidant function and reduced inflammatory damage	[1]
*Astragalus* polysaccharides	Basal diet supplemented with 100, 200, 400 mg/kg	Chongren hens	Improved production performance, egg quality, serum biochemical index and gut microbiota	[29]
gamma-irradiated *Astragalus* polysaccharides	Basal diet supplemented with 600 mg/kg	Ross-308 chicks	Improved growth performance and intestinal mucosal barrier function	[30,31]
*Astragalus membranaceus* (Fisch.) Bge. var. *mongholicus* (Bge.) Hsiao. polysaccharides	Basal diet supplemented with 200 ppm	Arbor Acres broiler chicks	Improved production performance, immune function and gut microbiota	[32]
*Astragalus membranaceus* and *Glycyrrhiza uralensis* polysaccharides	Basal diet supplemented with 300 mg/kg *Astragalus membranaceus* polysaccharides, 150 mg/kg *Glycyrrhiza uralensis* polysaccharides	Male Arbor Acres broiler chickens	Improved growth performance, intestinal health and gut microbiota	[17]
*Astragalus membranaceus* polysaccharide	Dissolved in 0.5 mL saline and injected into the air sac of the 7-day-old embryos through the wide end of the eggs	Eggs	Improved antioxidant activity, immune function, and liver and kidney functions	[14]
*Astragalus*	Cell supernatant 40 mg/L	Chicken embryo fibroblast Hailan white egg hen embryos	Attenuated chicken embryo fibroblast autophagy damage	[33]
alfalfa polysaccharide	Basal diet supplemented with 500 mg/kg	Arbor Acres broiler chicks (mixed sex)	Improved intestinal microbiota and systemic health	[34]
Caulis Spatholobi polysaccharides	Basal diet supplemented with 0.2, 0.4, 0.6%	Sanhuang cocks	Improved immunity, intestinal mucosal barrier function, and intestinal microbiota	[35]
*Glycyrrhiza uralensis* root polysaccharides	Administered by gavage at 3 different concentrations: 600, 450, 300 mg/kg for 14 d	Male Hy-Line brown chickens	Boosted immune function	[7]
Gan Cao (*Glycyrrhiza uralensis* Fisch) polysaccharides	Basal diet supplemented with 0.5, 1.0, 1.5%	Avian commercial female broilers	Enhanced growth performance, immune function, and gut microflora	[16]
Licorice *(Glycyrrhiza glabra)* polysaccharides	Basal diet supplemented with 200, 500, 1000, 1500 mg/kg	Male Arbor Acres broilers	Improved growth performance, serum antioxidant capacity, and biochemistry	[36]
*Atractylodes macrocephala* Koidz polysaccharides	Basal diet supplemented with 400 mg/kg	Magang goslings	Reduced oxidative stress and inflammatory response	[37]
*Atractylodes Macrocephala* Koidz Polysaccharide, purity 95%	Basal diet supplemented with 400 mg (kg body weight)	Specific pathogen-free goose	Promoted T lymphocytes activation and proliferation, restored the thymus cells morphology, alleviated immune suppression	[38]
*Artemisia ordosica* polysaccharide	Basal diet supplemented with 750 mg/kg	Arbor Acres broilers	Improved immune and antioxidative function	[5]
*Artemisia argyi* polysaccharide	Basal diet supplemented with 250, 500, 750, 1000 mg/kg	Arbor Acres broilers	Improved immune and antioxidative function	[39]
*Achyranthes bidentata* polysaccharides	Basal diet supplemented with 0, 0.02, 0.04%	Female Pekin ducks	Improved growth performance, immunity, antioxidant capacity, and meat quality	[9]
*Achyranthes bidentata* polysaccharides	Basal diet supplemented with 500 mg/kg	Female yellow-feathered broiler chickens	Promoted intestinal morphology, immune response, and gut microbiome	[40]
*Acanthopanax senticosus* polysaccharides	Basal diet supplemented with 0, 1, 2, 4 g/kg	Male Arbor Acres broiler chicks	Improved growth performance, immune function, antioxidation, and ileal microbial populations	[3]
ginseng polysaccharides	Basal diet supplemented with 200 g/t	Xuefeng blackbone chickens	Improved intestinal morphology and microbiota composition	[41]
*Camellia oleifera* cake polysaccharides	Basal diet supplemented with 0, 200, 800 mg/kg	Lingnan yellow broiler	Improved growth performance, carcass traits, meat quality, blood profile, and caecum microorganisms	[42]
Yingshan Yunwu tea polysaccharides	Basal diet supplemented with 200, 400, 800 mg/kg	Chongren chickens	Improved meat quality, immune status and intestinal microflora	[43]
*Codonopsis pilosula* polysaccharide	Orally treated with CPPS (3 mg per feather), pCPPS (2.5 mg per feather) for 3 consecutive days	Ducklings	Promoted immune ability	[10]
*Platycodon grandifloras* polysaccharides	Cell supernatant incubated in DMEM containing 200 μg/mL	Chicken embryo fibroblast cell lines	Attenuated chicken embryo fibroblast cell lines mitochondrial damage	[44]
*Lycium barbarum* polysaccharides	Basal diet supplemented with 1000, 2000, 4000 mg/kg	Male Arbor Acres broiler chicks	Improved growth performance, digestive enzyme activities, antioxidant status, and immunity	[45]
Mulberry leaf polysaccharide, purity 95%	Oral: at doses of 8, 4, 2 mg continuously for 7 d	Male chicks	Enhanced the respiratory mucosal barrier immune response	[46]
*Paulownia tomentosa* flower polysaccharide	Oral: at different doses of 50, 25, 12.5, 6.25, 3.125 mg/kg for 3 successive days	White Roman chickens	Enhanced immunological activity	[47]
*Paulownia fortunei* flowers polysaccharide	Subcutaneously injected in the neck with 0.25 mL at concentrations of 40, 20, 10 mg/mL for 3 successive days	Specific pathogen-free chickens	Enhanced cellular and humoral immunity	[48]
selenizing *Schisandra chinensis* polysaccharide	Cell supernatant SCP and sSCP were respectively twofold diluted from 1600 μg/mL to 1.563 μg/mL with DMEM	Chicken embryo hepatocytes	Attenuated hepatocyte oxidative damage	[49]
*Aloe vera* polysaccharides	Oral: at the dose rates of 100, 200, 300 mg/kg body weight	Broiler chicks	Enhanced immunotherapeutic effects and anti-coccidial	[12]
*Amomum longiligulare* T.L. Wu fruits polysaccharide	Injection: 30 mg ALP1 (glucose) with 0.25 mL normal saline; 30 mg ALP2 (glucose, glucuronic acid and galacturonic acid with the ratio of 91.43:5.23:3.34.) with 0.25 mL normal saline	Chickens	Promoted bursa of Fabricius immune ability	[11]
Taishan *Pinus massoniana* pollen polysaccharides	Orally administered with 10, 20, 40 mg/mL (0.2 mL/chicken) for 21 consecutive days	Specific pathogen–free chickens	Enhanced intestinal mucosal immunity and intestinal villi development	[50]
Taishan *Pinus Massoniana* pollen polysaccharide	Oral: received 5.0 mg for 7 consecutive days	Specific pathogen-free chickens	Enhanced antiviral and antitumor activity	[13]
Taishan *Pinus massoniana* pollen polysaccharide	Cell supernatant concentrations, including 0, 12.5, 50, 100, 200, 400, 800, 1600 μg/mL	Chicken peripheral blood lymphocytes	Enhanced host immune response	[51]
Yupingfeng polysaccharides	Basal diet supplemented with 0.5, 1, 2, 4 g/kg	Qingyuan partridge chickens	Enhanced growth performance and small intestinal digestion and absorption	[52]

## Data Availability

All data referred to in the manuscript are already published.

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
