# Peer review of "Progress of Studies on Plant-Derived Polysaccharides Affecting Intestinal Barrier Function in Poultry"

_animals, 2022, doi:10.3390/ani12223205_

Round 1
Reviewer 1 Report (Previous Reviewer 2)
The manuscript is very interesting and fits in the scope of the Animals journal. It provides information on the effects of plant polysaccharides on poultry gut structure and function. The authors have divided the text into four chapters. The division of the text into subsections is appropriate.
I have thoroughly reviewed the manuscript. I believe the authors addressed all the issues raised by the reviewer satisfactorily. I do not have any comments or suggestions before its acceptance to be published in Animals, and I recommend this paper for publication.
Author Response
Thank you for your approval of this manuscript, which has been further refined.
Reviewer 2 Report (New Reviewer)
The paper is a interesting review of a theme, but it does not propose any kind of use of these components in poultry nutrition. in my opinion, it is just a review and it does explain the real effects. There are especulations that could be interesting, but we need more propositive information.

Author Response
26/Oct/2022
Dear Reviewer:
Thank you for your letter and the comments concerning our manuscript (Manuscript ID: animals-1945279). The manuscript has been revised carefully according to the comments of the reviewers. Please find the revised version of our manuscript entitled ‘Recent Progress of Studies on Plant-derived Polysaccharides Affecting Intestinal Barrier Function in Poultry’. The detailed revision contents in the manuscript and the response to the reviewers’ comments are listed as follows:
Commented [A1]: Better use alphabetical order
Responses: We have reordered in the revised manuscript (Line 23-24).
Commented [A2]: Change "and so on" for other term more adequate. It seems very casual.
Responses: We have revised it in the revised manuscript (Line 34).
Commented [A3]: Facilitate it is not the activity of a carbohydrate, change for a term more correlated to the real action of these substances.
Responses: Thanks for the valuable advice, we have revised it in the revised manuscript (Line 67).
Commented [A4]: Are You affirming that these components can act directly in digestive and absorptive functions in small intestine? This was my comprehension. I believe this is the key point of the paper and must be well explained and justified.
Responses: Up to now, most reports on plant polysaccharides have not found that plant polysaccharides directly act on the digestion and absorption function of small intestine. In our manuscript, we added the explanation that plant polysaccharides indirectly promote the digestion and absorption function of small intestine. Moreover, we found, from some reports, that other polysaccharides are not digested in the upper gastrointestinal tract, but they can retain water, promote peristalsis, encourage satiety feelings, and retard the rate of emptying in some reports (Line 316-323, 413-414).
Commented [A5]: This table is very useful, but you should explain why you showed all of these results. It is a compilation. I understand that it could be a summary of the results but you could organize according to the main function of each component used. I understand that it was not a simply test without a clear objective.
Responses: Thanks for the valuable advice, we have added a description in the revised manuscript (Line 90-92, Table 1).
Commented [A6]: I read all of this section and you do not explain the reasons why these substances affect microbial barrier. If you cite a table and the reason of changing microbial composition, it will be more applied.
Responses: Thanks for your valuable advice, we have added a description in the revised manuscript (Line 316-323).
Commented [A7]: Do not use abbreviations without previously define it
Responses: We have revised them in the revised manuscript (Line 259-260).
Commented [A8]: The same conclusion, it is a description without describing a reason for the effects expected.
Responses: At the end of this section, we briefly summarize three potential reasons (Line 406-417).
Commented [A9]: mL
Responses: We have revised it in the revised manuscript (Line 363).
Commented [A10]: It is a description, without explanation
Responses: At the end of this section, we add possible explanations (Line 536-550).
Commented [A11]: But is it not possible to speculate based on other animal species observations?
Responses: This is a very meaningful question. As you say, it is possible to infer from other species. In our manuscript, the effects of existing plant polysaccharides on the intestinal barrier function of poultry were summarized.
Commented [A12]: This is not 100% adequate without a functional analysis
Responses: This is a worthy question. There is no functional analysis in the existing reports, which needs to be further studied to fill the gap in the intestinal digestion and absorption of plant polysaccharides in poultry.
Thank you for your consideration. I look forward to hearing from you.
Sincerely,
Binlin Shi
Reviewer 3 Report (New Reviewer)
Overall, the review approaches an interesting and very up-to-date issue of Animals. The manuscript is quite comprehensive but unfortunately it is unsatisfactory at this stage. There are some major deficiencies which I need to raise. My concerns address the content, as well as the linguistic quality of the manuscript. In my opinion, the manuscript should not be considered for publication in the present form.
General comments:
1/ The manuscript requires profound English proofing. Many sentences are very long, complicated and chaotic. Please correct.
2/ There is no information on the methodology of data search for the review.
2/ Both, the simple summary and abstract, contain terms like “chemical barriers, physical barriers” which probable won’t be clear to general audience.
3/ The manuscript is focused on various aspect of intestine barrier in poultry. However, it does not refer to the effect of polysaccharides on gut contractility which in turn can affect e.g. intestine microbiota. Can you extend the scope of the manuscript and include data on polysaccharide impact on gut motility?
4/ The is no data on polysaccharide-rich plants (sources) which could be used as feed supplements on general poultry performance.
5/ The figures seem to be of low quality.
Detailed comments:
L 75: it is too general to state that polysaccharides have no toxicity. It is only the matter of dose. Please re-phrase the statement.
L 134: Taking into account local secretion in the intestine, it is hard to agree that gastric acid contributes largely to chemical barrier in the small intestine.
L 143 (also 474, 482, 502): “intestinal flora” – it is not a correct term
Table 1: “Reference contents” – This column refers sometimes to the object of the study and sometimes to the results (conclusions) – please unify.
L 507-510: it is not clear what the authors wish to express. “To inhale into the body”?
L 510-512: “Of course, it was also reported that plant-derived polysaccharides 510 mediated circadian rhythms and related psychiatric disorders through intestinal microbiota” – the sentence is out of the contest.
Author Response
26/Oct/2022
Dear Reviewer:
Thank you for your letter and the comments concerning our manuscript (Manuscript ID: animals-1945279). The manuscript has been revised carefully according to the comments of the reviewers. Please find the revised version of our manuscript entitled ‘Recent Progress of Studies on Plant-derived Polysaccharides Affecting Intestinal Barrier Function in Poultry’. The detailed revision contents in the manuscript and the response to the reviewers’ comments are listed as follows:
General comments:
1/ The manuscript requires profound English proofing. Many sentences are very long, complicated and chaotic. Please correct.
Responses: We have corrected it in the revised manuscript.
2/ There is no information on the methodology of data search for the review.
Responses: From Table 1, we can intuitively see the plants as the source of plant-derived poly-saccharides, application forms, dose range, experiment objects, and their main functions in poultry production.
2/ Both, the simple summary and abstract, contain terms like “chemical barriers, physical barriers” which probable won’t be clear to general audience.
Responses: We have revised it in the revised manuscript (Line12-13, 19).
3/ The manuscript is focused on various aspect of intestine barrier in poultry. However, it does not refer to the effect of polysaccharides on gut contractility which in turn can affect e.g. intestine microbiota. Can you extend the scope of the manuscript and include data on polysaccharide impact on gut motility?
Responses: Thank you for your valuable advice. Unfortunately, there are no data on the effects of plant polysaccharides on intestinal motility in poultry. Previous reports have suggested that marine algae polysaccharides are not digested in the upper gastrointestinal tract, but they can retain water, promote peristalsis, encourage satiety feelings, and retard the rate of emptying. This has been found in human intestines but has not been reported in poultry intestines.
4/ The is no data on polysaccharide-rich plants (sources) which could be used as feed supplements on general poultry performance.
Responses: In the field of poultry research, our manuscript summarizes many types of polysaccharides such as Astragalus, Glycyrrhiza uralensis, Mulberry, Aloe vera, Pinus massoniana, Codonopsis pilosula, Paulownia tomentosa, Amomum longiligulare T.L. and Artemisia ordosica, etc. These plant polysaccharides will provide a reliable reference for future applications in poultry production.
5/ The figures seem to be of low quality.
Responses: The figures are mainly quoted from others, which can directly reflect the relevant content of this paper.
Detailed comments:
- L 75: it is too general to state that polysaccharides have no toxicity. It is only the matter of dose. Please re-phrase the statement.
Responses: We have revised it in the revised manuscript (Line 31).
- L 134: Taking into account local secretion in the intestine, it is hard to agree that gastric acid contributes largely to chemical barrier in the small intestine.
Responses: We have revised it in the revised manuscript.
- L 143 (also 474, 482, 502): “intestinal flora” – it is not a correct term
Responses: We have revised it in the revised manuscript (Line 83, 268, 276, 311-312, 324,).
- Table 1: “Reference contents” – This column refers sometimes to the object of the study and sometimes to the results (conclusions) – please unify.
Responses: We have revised it in the revised manuscript (Table 1).
- L 507-510: it is not clear what the authors wish to express. “To inhale into the body”?
Responses: We have revised it in the revised manuscript (Line 324).
- L 510-512: “Of course, it was also reported that plant-derived polysaccharides mediated circadian rhythms and related psychiatric disorders through intestinal microbiota” – the sentence is out of the contest.
Responses: We have deleted it in the revised manuscript.
Thank you for your consideration. I look forward to hearing from you.
Sincerely,
Binlin Shi
Round 2
Reviewer 2 Report (New Reviewer)
There are some points that must be reviewed, specially insome sentences that are confuse about the use of comma or typing errors.
I suggest to delete the word "recent" from the title. It is recent now, but i
Table 1 is very long and difficult for interpretation. I really prefer that the authors organize better the results, in classes of produts or results. The authors should prefer to cite results that are similar in their conclusion to be easier to understand the purpose for how much data exposed in just one table.

Author Response
13/Nov/2022
Dear Reviewer,
Thank you for your letter and the comments concerning our manuscript (Manuscript ID: animals-1945279). The manuscript has been revised carefully according to the comments of the reviewers. Please find the revised version of our manuscript entitled ‘Progress of Studies on Plant-derived Polysaccharides Affecting Intestinal Barrier Function in Poultry’. The detailed revision contents in the manuscript and the response to the reviewers’ comments are listed as follows:
Reviewer 2: Comments and responses
Comment: There are some points that must be reviewed, specially in some sentences that are confuse about the use of comma or typing errors.
Responses: We have revised them in the revised manuscript.
Comment: I suggest to delete the word "recent" from the title. It is recent now, but i
Responses: We have deleted it in the revised manuscript (Line 2).
Comment: Table 1 is very long and difficult for interpretation. I really prefer that the authors organize better the results, in classes of products or results. The authors should prefer to cite results that are similar in their conclusion to be easier to understand the purpose for how much data exposed in just one table.
Responses: Table 1 has been sorted according to the type of products, and the references in it have been cited for many times in the subsequent paragraphs.
Commented [A1]: This comment is too general. In my opinion is dispensable
Responses: We have deleted it in the revised manuscript.
Commented [A2]: This phrase seemed to me out of the context of the paragraph. I recommend re-moving this paragraph, rewriting and directing to the next item (3.2)
Responses: We have deleted it in the revised manuscript.
Commented [A3]: who was the author who made this statement?
I understand that it is better to clearly cite the author(s)
Responses: We have added it in the revised manuscript (Line 312).
Commented [A4]: serotype
Responses: We have revised it in the revised manuscript (Line 408).
Commented [A5]: Many?
Responses: We have revised it in the revised manuscript (Line 818).
Thank you for your consideration. I look forward to hearing from you.
Sincerely,
Binlin Shi
Reviewer 3 Report (New Reviewer)
Thanks for addrsing my suggestions.
When I wrote "missing methodology", I meant no information how the data were collected (which Data Bases, Search tools, etc). I realiza the Table 1 summerized nicely collected data.
Author Response
13/Nov/2022
Dear Reviewer,
Thank you for your letter and the comments concerning our manuscript (Manuscript ID: animals-1945279). The manuscript has been revised carefully according to the comments of the reviewers. Please find the revised version of our manuscript entitled ‘Progress of Studies on Plant-derived Polysaccharides Affecting Intestinal Barrier Function in Poultry’. The detailed revision contents in the manuscript and the response to the reviewers’ comments are listed as follows:
Reviewer 3: Comments and responses
Comments When I wrote "missing methodology", I meant no information how the data were collected (which Data Bases, Search tools, etc). I realiza the Table 1 summerized nicely collected data.
Responses: Thanks again for your question. The data is collected from different references and involves multiple databases (PubMed, Web of science, Elsevier, Springer…) in our manuscript.
Thank you for your consideration. I look forward to hearing from you.
Sincerely,
Binlin Shi
This manuscript is a resubmission of an earlier submission. The following is a list of the peer review reports and author responses from that submission.
Round 1
Reviewer 1 Report
The authors of this review have compiled many references on a range of sources of plant-derived polysaccharides that have been used to improve the intestinal health of poultry.
The plant sources are listed in their Table 1 however, the scientific names at a species level are not always included: this is particularly so for the genus Astragalus. The different species of Astragalus have different active compounds (eg Li et al. A Review of Recent Research Progress on the Astragalus Genus Molecules 2014, 19, 18850-18880; doi:10.3390/molecules191118850).
The parts of the plant used or whether the treatment applied was an extract is not included in the table.
It is also not clear in the table what the difference between “oral” or “dietary” administration is and little detail is given in the text to explain this.
General comments on text
A good review article should not only cover the relevant literature but should critically evaluate the experimental evidence on the review topic. It should not be just a descriptive statement of the results reported in experimental papers but should try to compare information from the literature on a topic.
Authors should try to provide the reader of a review with their interpretation of experiments and some explanation – people reading a review should not have to read every paper for themselves.
Experiments should be compared - were similar methods used; what amounts of polysaccharide sources were used; what was the method of application of the polysaccharide; does method of application or prior treatment of the polysaccharide source affect the results? These are examples of the questions that readers may want to know.
If the authors of a specific paper on poultry have not analysed the polysaccharide source they used, are there other research papers in which analyses are reported and which could be used to shed light on the possible mechanisms of action of the treatment applied.
As many of the papers use whole plants tissues in the treatments applied, perhaps mention should be made, where relevant, of possible active components besides polysaccharides that may be present in the tissues.
I note that there are at least 13 references listed out of 67 that are in Chinese. As this review is being written in English, I assume the authors would like to enlighten the English reading public, who probably cannot read Chinese, of the work being done. To do so requires more information than the mostly simple statements on the results of research papers.
English grammar
I have not listed all the grammatical errors or sentences that are difficult to understand.
A few examples are:
Line 38 – “about the application study”
Lines 59 to 60 – “… the review object is not for certain livestock or poultry…”
Lines 281 – 284 – the polysaccharide was not “supplemented” with gamma-irradiation.
Section 3.2 starts with a very long paragraph (Lines 85 – 132) which is difficult for a reader to read. It should be broken into shorter paragraphs. Especially there where there is a change of subject to fatty acids.
Other paragraphs in the text are very long and could do with being broken into smaller paragraphs.
References
I have not checked to see if any in the text are not in the reference list or that they are correctly quoted. I also note there is some inconsistency in the formatting of references.
Reviewer 2 Report
The manuscript is very interesting and fits in the scope of the Animals journal. It provides information on the effects of plant polysaccharides on poultry gut structure and function.
Detailed comments:
1. Introduction: paragraphs 1 and 2 are not related, write a few sentences on the relationship of polysaccharides to the gut to support the topic; poultry are also livestock
2. Source and structure of plant-derived polysaccharides:
Table 1: which chickens - broilers or layers?; chicken broilers, duck broilers, goose broilers, turkey broilers? - please specify
3. Effects of plant-derived polysaccharides on intestinal barrier function in poultry: the chapter is rightly divided into subsections, making the manuscript clearer, but I have a few comments
3.1. Intestinal health: The authors emphasise that in poultry, intestinal health affects the functioning of the body, is it different in other animal species or in humans?; I think this chapter is superfluous, it does not add anything concrete to the content, it should be expanded or the information it contains should be moved to chapter 3.2.
3.2. Effects of plant-derived polysaccharides on intestinal microbial barrier in poultry : I have no comments
3.3. Effects of plant-derived polysaccharides on intestinal chemical barrier in poultry: I have no comments
3.4. Effects of plant-derived polysaccharides on intestinal physical barrier in poultry: I have no comments
3.5. Effects of plant-derived polysaccharides on intestinal immune barrier in poultry: line 442 „Caspase” and line 447,448 „Chromium” – please write in lower case
4. Conclusions and perspectives: there is a lack of defining perspectives: does such research have a future? what can they be used for? what practical dimension do they have? can they be applied to other animal species?
5. References: very topical